# Consumption of Policosanol (Raydel^®^) Improves Hepatic, Renal, and Reproductive Functions in Zebrafish: In Vivo Comparison Study among Cuban, Chinese, and American Policosanol

**DOI:** 10.3390/ph17010066

**Published:** 2023-12-30

**Authors:** Kyung-Hyun Cho, Ji-Eun Kim, Hyo-Seon Nam, Seung-Hee Baek, Ashutosh Bahuguna

**Affiliations:** Raydel Research Institute, Medical Innovation Complex, Dong-gu, Daegu 41061, Republic of Korea; ths01035@raydel.co.kr (J.-E.K.); sun91120@raydel.co.kr (H.-S.N.); shbaek@raydel.co.kr (S.-H.B.); ashubahuguna@raydel.co.kr (A.B.)

**Keywords:** policosanol, high-cholesterol diet (HCD), high-density lipoproteins (HDL), apolipoproteinA-I (apoA-I), inflammation, interleukin-6, liver, kidney, ovary, testis

## Abstract

The current study compared three policosanols from Cuba (sugarcane, Raydel^®^, policosanol (1), China (rice bran, Shaanxi, policosanol (2), and the USA (sugarcane, Lesstanol^®^, policosanol (3) in the treatment of dyslipidemia and protection of the liver, ovary, and testis in hypercholesterolemic zebrafish. After twelve weeks of supplementation of each policosanol (PCO, final 0.1% in diet, *w*/*w*) with a high cholesterol diet (HCD, final 4%, *w*/*w*), the Raydel policosanol (PCO1) group showed the highest survivability, approximately 89%. In contrast, Shaanxi policosanol (PCO2) and Lesstanol policosanol (PCO3) produced 73% and 87% survivability, respectively, while the HCD alone group showed 75% survivability. In the 12th week, the PCO1 group demonstrated the most modest increase in body weight along with significantly lower levels of total cholesterol (TC) and triglycerides (TG) in comparison to the HCD control group. Additionally, the PCO1 group exhibited the highest proportion of high-density lipoprotein (HDL)-cholesterol within TC. Notably, the PCO1 group displayed the lowest level of aspartate aminotransferase and alanine aminotransferase, minimal infiltration of inflammatory cells, reduced interleukin (IL)-6 production in the liver, a notable decline in reactive oxygen species (ROS) generation and mitigated fatty liver changes. HCD supplementation induced impairment of kidney morphology with the greatest extent of ROS production and apoptosis. On the other hand, the PCO 1 group showed a remarkably improved morphology with the least ROS generation and apoptosis. Within the ovarian context, the PCO1 group exhibited the most substantial presence of mature vitellogenic oocytes, accompanied by minimal levels of ROS and apoptosis. Similarly, in the testicular domain, the PCO1 group showcased optimal morphology for spermatogenesis, characterized by the least interstitial area and diminished production of ROS in testicular cells. At week 8, the PCO1 group showed the highest egg-laying ability, with around 244 eggs produced per mating. In contrast, the HCD alone, PCO2, and PCO3 groups showed significantly lower egg-laying ability (49, 59, and 86 eggs, respectively). The embryos from the PCO1 group exhibited the highest survivability with the fastest swimming ability and developmental speed. These results suggest that PCO1 consumption significantly enhanced the reproduction system, egg-laying ability, and embryo survivability. In conclusion, among the three policosanols, Cuban (Raydel^®^) policosanol had the strongest effect on survivability, improving dyslipidemia, liver protection, kidney, ovary, and testis with a restoration of the cell morphology, and the least ROS production and apoptosis-induced by HCD supplementation.

## 1. Introduction

Raising the serum HDL-C and lowering the TC and LDL-C is an excellent strategy to maintain healthy longevity by suppressing dyslipidemia and hypertension [1,2]. Dyslipidemia, particularly low HDL-C, is a systemic risk factor for many multi-organ failures, such as acute pancreatitis [3] and sepsis [4]. In addition, low HDL-C is also associated with reproduction disorders, such as unexplained female infertility [5] and erectile dysfunction [6], as well as cardiovascular risk factors. Moreover, in non-alcoholic fatty liver disease (NAFLD), reduced levels of HDL-C may serve as an indicator for the onset of liver cancer in conjunction with hepatic fibrosis [7,8]. Furthermore, low HDL-C levels are also associated with the induction of chronic renal disorder, i.e., low HDL-C is frequently connected with decreasing glomerular filtration rate (eGFR) [9].

Among the many functional foods available, policosanol (PCO) products are in strong demand, with several beneficial activities to prevent dyslipidemia and hypertension [10,11]. The global policosanol market is USD 299.8 million in 2023 and is expected to reach USD 588.3 million in 2033, with a forecasted 6.6% annual growth rate from 2023 to 2033 (https://www.futuremarketinsights.com/reports/policosanol-market, accessed on 16 September 2023). 

Many policosanol products, approximately 25 brands with a 5–40 mg dosage per day, have been developed worldwide as functional foods that commonly claim to lower the blood cholesterol (total cholesterol, TC) level and raise the high-density lipoproteins-cholesterol (HDL-C) level. Since the first policosanol was purified from sugarcane in 1993 [12], various policosanol products have been isolated from different natural sources and mixed in the global market with some confusion of their origin, such as rice bran [13] and insects [14]. Furthermore, the different sources and brands of policosanol exhibited different ingredient compositions of long-chain aliphatic alcohols, such as octacosanol content. For example, Cuban policosanol (Raydel) had an approximate 70% octacosanol content [15], while American policosanol (Lesstanol^®^, Octa60) contained 61% octacosanol [16], and Chinese policosanols contained various octacosanol contents (11–95%) [15]. 

There has been considerable debate regarding the efficacy of policosanol with controversial data because many similar policosanol products have been marketed as copycats, but clinical data and adverse effects are lacking, resulting in customer confusion. A policosanol consisting of 6.2% hexacosanol, 12.6% triacontanol, and 62.9% octacosanol [17,18], similar to Lesstanol^®^, was reported to possess potent cholesterol-lowering properties with the prevention of lipoprotein peroxidation [17]. The same policosanol, like Lesstanol^®^, was considered a promising agent in preventing and treating cardiovascular disease (CVD) by lowering the TC and LDL-C and raising HDL-C [18]. On the other hand, Lesstanol^60^, consisting of 61.2% octacosanol, 16.8% triacontanol, and 6.5% hexacosanol, showed no potential to reduce the CVD risk without improving the serum lipid profile in heterozygous LDL-receptor deficient mice and not de novo production of apoA-I [19]. In contrast, Cuban policosanol (Raydel^®^), consisting of 70.4% octacosanol, 14.1% triacontanol, and 3.8% hexacosanol, showed potent efficacy in treating dyslipidemia in hyperlipidemic animal models and pre-hypertension in human clinical studies [20,21]. On the other hand, Chinese policosanols from sugar cane wax (11% or 42% octacosanol) and rice bran (95% octacosanol) did not show adequate antioxidant and anti-glycation activity in an in vitro study [15], and no improvement in dyslipidemia in an in vivo study [20]. These studies suggest that the specific compositions of individual aliphatic alcohols might be crucial to influencing the in vitro rHDL formation ability and the in vivo efficacy in animals and humans, depending on the origin of plant sources and countries. 

Regarding plant sources, sugar cane wax alcohol (policosanol) exhibits different properties from rice bran policosanol [15,16,20]. Two rice bran extracts from the *Indica* and *Japonica* rice cultivars showed different phytochemical contents and cytoprotective effects [22], suggesting that different sub-species of rice bran might contain policosanol with different properties. Moreover, despite the reported antidiabetic effect of Egyptian rice bran policosanol [23] and attenuation of thrombosis by Malaysian rice bran policosanol [24], there is little information comparing sugarcane policosanol and rice bran policosanol in the same in vitro and in vivo study. 

The studies carried out by our research group compared three distinct Chinese policosanols, Xi’an Natural sugarcane, Xi’an Realin sugarcane, and Shaanxi rice bran 980, and Cuban sugarcane policosanol, regarding their antioxidant, antiglycation, and anti-inflammatory activities via reconstituted HDL synthesis [15]. Moreover, in the same context, Cuban policosanol (Raydel), Chinese (Shaanxi rice bran 400), and American policosanol (Garuda, Lesstanol) were compared using in vitro assays via rHDL synthesis [16]. On the other hand, there are no reports comparing the in vivo efficacy of the three policosanols in a hyperlipidemic animal model to provide useful information on the appropriate choice for consumers to select the policosanol product. 

Owing to the protective role of HDL in a multi-organ system, it is worth assessing whether policosanol supplementation improves dyslipidemia, inflammation, organ damage, and infertility with embryonal defects by raising the HDL-C and HDL quality. The current study evaluated the in vivo efficacy of the three policosanols from Cuba, China, and the USA in treating dyslipidemia and protecting the liver, kidney, ovary, and testis in a hyperlipidemic zebrafish model. Zebrafish are prone to the induction of hyperlipidemia, while rats and mice are resistant to the induction of hyperlipidemia upon HCD consumption. Zebrafish have many similar organs to humans with blood, kidney, liver, heart, and brain. Therefore, zebrafish have been a popular model system for studying human diseases, particularly dyslipidemia, inflammation, and muti-organ failure [25,26]. 

Among the different in vivo models, zebrafish catches huge attention for preclinical [27] and toxicological studies [28] due to its higher homology with the human genome [29]. Numerous reports suggest that assessing the preclinical efficacy of small molecules in zebrafish has facilitated their transition into clinical trials [27], highlighting the suitability of zebrafish as an exceptional model for preclinical investigations. Notably, the lipid metabolism process in zebrafish closely resembles that in humans, as zebrafish possess critical receptors, lipoproteins, apolipoproteins, and various enzymes essential for lipoprotein metabolism [30]. In contrast to mice, zebrafish demonstrate the presence of cholesteryl ester transfer protein (CETP), an important protein in lipid metabolism [30]. These distinctive qualities make zebrafish an outstanding model for research on lipoprotein and the related field.

Based on the literature findings mentioned earlier [10,11,20,21], policosanols have demonstrated effectiveness against various ailments. However, comparative studies are scarce among different policosanols, specifically concerning dyslipidemia. To expand our understanding, it is valuable to investigate the impact of policosanol supplementation against dyslipidemia-induced inflammation, organ damage, and infertility by elevating HDL-C levels and improving HDL quality. With this consideration, the present investigation was initiated to assess the in vivo efficacy of three distinct policosanols sourced from Cuba, China, and the USA in addressing dyslipidemia and safeguarding essential organs (liver, kidney, ovary, and testis) employing the hyperlipidemic zebrafish model.

## 2. Results

### 2.1. Change in the Mortality and Body Weight

After twelve weeks of supplementation, the ND group and HCD group (final 4% cholesterol) showed 89% and 74% survivability, respectively, implying that the consumption of a HCD was a risk factor for a healthy lifespan and that 4% cholesterol supplementation (HCD group) induced acute death via the induction of hyperlipidemia and hyperinflammation (Figure 1A). Under HCD consumption, the PCO1 (Raydel) group showed 89% survivability, while the PCO2 (Shaanxi) and PCO3 (Garuda) groups exhibited lower survivability (73% and 87%, respectively).

The ND and HCD control group exhibited a 1.9-fold and 2.6-fold increase in body weight (BW) at week twelve compared to the initial BW (week 0), as shown in Figure 1B. Under HCD, however, the PCO1 group showed a 2.0-fold increase in BW, while the PCO2 and PCO3 groups exhibited a 2.4-fold increase in BW. At week 12, among HCD groups, the PCO1 group showed the lowest BW, 585 ± 28 mg, which was a 22% lower BW than the HCD alone group (*p* = 0.006). On the other hand, the PCO2 and PCO3 groups showed ~689 ± 33 mg, which was a 7–8% lower BW than the HCD alone group without significance. These results indicate that the Cuban policosanol (PCO1, Raydel) group exhibited the highest survivability and the lowest increase in BW during the 12 weeks of consumption under HCD supplementation. 

### 2.2. Changes in Blood Lipid Profiles

Following twelve weeks of consumption, the HCD alone group exhibited 1.8-fold and 2.0-fold higher TC (Figure 2A) and TG (Figure 2B) levels, respectively, than those of the ND group, implying that the 4% cholesterol supplementation elevated the blood TC and TG levels by approximately 861 ± 14 mg/dL and 954 ± 42 mg/dL, respectively, as shown in Figure 2. On the contrary, among the HCD groups, the PCO1 group showed the lowest TC (454 ± 8 mg/dL) and TG (662 ± 11 mg/dL) levels, which were up to 48% and 31% lower than the HCD alone group, respectively; while the PCO2 and PCO3 groups showed similar TC levels, ~833–888 mg/dL, and TG levels, ~898–922 mg/dL with the HCD alone group. The Chinese policosanol (PCO2) and American policosanol (PCO3) groups exhibited almost no blood lipid lowering activities (TC and TG) compared to the Cuban policosanol (PCO1) group. 

As depicted in Figure 2C,D, the HCD alone group showed the lowest HDL-C, ~109 ± 2 mg/dL, and HDL-C/TC (%), ~12.7%, while the ND group showed a 1.3-fold and 2.6-fold higher HDL-C and HDL-C/TC (%), respectively, than the HCD group. Among the HCD groups, however, the PCO1 group showed a 2.2-fold higher HDL-C level (236 ± 4 mg/dL) and a 4.1-fold higher HDL-C/TC (%) than the HCD alone group. In contrast, the PCO2 group showed weaker activity to raise the HDL-C around 149 ± 3 mg/dL, which was 1.4-fold higher than the HCD alone group, and almost no activity to raise HDL-C/TC (%) around 16.8%. Although the PCO3 group showed 2.5-fold higher HDL-C than the HCD-alone group, the HDL-C/TC (%) was 2.5-fold higher than the HCD-alone group due to the elevation of TC. These results suggest that Cuban policosanol (PCO1) consumption had the most desirable lipid profile to exhibit the lowest TC and TG with the highest ratio of HDL-C in TC among the HCD groups. By contrast, Chinese policosanol (PCO2) and American policosanol (PCO2) consumption caused an abrupt elevation of the TC and TG levels and a decrease in HDL-C and HDL-C/TC (%). 

### 2.3. Liver Function Parameters

The HCD alone group exhibited 11.4-fold (*p* < 0.001) and 16.5-fold (*p* < 0.001) higher blood AST and ALT levels, respectively, than those of the ND alone group after twelve weeks of consumption (Figure 3), indicating that cholesterol (final 4%) supplementation caused severe injury to the liver and muscle. On the other hand, the PCO1 group exhibited 85% and 90% lower AST and ALT levels (*p* < 0.001) than the HCD group, which were the lowest levels among the HCD groups. In contrast, the PCO2 group showed 49% and 33% lower levels than the HCD group, which were the highest levels among the three policosanol groups. Interestingly, the PCO3 group displayed a 74% and 33% decrease in the AST and ALT levels, respectively, than the HCD group, implying that there might be more liver-specific damage than muscle damage during the twelve weeks of consumption. These results suggest that Cuban policosanol supplementation induced the strongest protection against muscle and liver damage from hyperlipidemia, while Chinese and American policosanol offered much weaker protection against muscle and liver damage (Figure 3).

### 2.4. Analysis of the Liver Tissue (H&E Staining and Oil-Red O Staining)

Liver tissue analysis using H&E staining was performed to monitor the hepatological changes in the different groups, as shown in Figure 4. The HCD control group (photographs b1 and b2) displayed the highest H&E-stained area (Figure 4A), with accumulated and massive neutrophil infiltration that was 3.6-fold higher (*p* < 0.001) than the ND alone group (photographs a1 and a2). In contrast, the PCO1 group (photographs c1 and c2) inhibited the HCD-induced infiltration of neutrophils with up to ~74% lower (*p* < 0.001) neutrophil counts than that of the HCD alone group (Figure 4B). On the other hand, the PCO2 and PCO3 groups did not alleviate the infiltration of neutrophils caused by the HCD (Figure 4B). The H&E-stained area monitored for the PCO2 and PCO3 groups showed similarity to the HCD alone group (photographs d1, d2, e1, and e2) (Figure 4). The PCO2 and PCO3 groups showed even higher neutrophil counts than the HCD control group, but the difference was not significant. 

Oil red O staining of the hepatic section revealed a 7.0-fold higher oil red stained area in the HCD alone group (photograph b3) than the ND alone group (photo a3), as depicted in Figure 4A, implying a severe fatty liver change in response to cholesterol supplementation. Among the HCD groups, however, the PCO1 group (photo c3) prevented the fatty liver changes with a marked 90% reduction (*p* < 0.001) of the oil-red stained area than the HCD alone group (Figure 4B). On the contrary, the PCO2 (photograph d3) and PCO3 group (photo e3) showed a weaker effect, ~29% and 21% decrease (*p* < 0.05), compared to the HCD supplementation-induced fatty liver changes. 

### 2.5. Interleukin-6 and ROS Production

As shown in Figure 5A, the immunohistochemistry (IHC) of hepatic tissue revealed that the HCD alone group (photographs b1 and b2) showed the highest IL-6 production, as evidenced by the 16.1-fold higher (*p* < 0.001) IHC stained area (Figure 5B) than that of the ND alone group (photographs a1 and a2), implying the impact of cholesterol supplementation for twelve weeks on more IL-6 generation in hepatic tissue. The PCO1 group (photographs c1 and c2) exhibited the smallest IL-6-stained area (4.7 ± 0.7%), which was 85% smaller (*p* < 0.001) than that of the HCD alone, indicating the impact of PCO1 to mitigate the HCD induced IL-6 production. Similarly, the PCO2 group (photographs d1 and d2) also reduced IL-6 generation (13.7 ± 1.1%) with a 58% lower level (*p* < 0.01) than the HCD alone. In contrast, the PCO3 group showed relatively higher IL-6 production, 23.1 ± 1.6%, which was 29% lower than that of the HCD group (Figure 5B). Although the PCO3 group showed a lower IL-6 level than the HCD alone group, it was the highest IL-6 level among the three policosanol groups, similar to the serum ALT level. 

DHE staining showed that the HCD alone group (photograph b3) has a 3.6-fold higher ROS generation than that of the ND group (photograph a3), highlighting the impact of cholesterol consumption on ROS generation. Among the HCD-supplemented groups, the PCO1 (photo c3) and PCO2 groups showed 75% and 52% lower ROS generation than the HCD alone group, as quantified by the DHE stained area (Figure 5B). By contrast, the PCO3 group (photograph f1) had no inhibitory effect against the HCD-induced ROS generation. Furthermore, 32% more ROS generation was observed in the PCO3 group compared to the HCD alone group, indicating severe ROS generation and fatty liver changes. Hence, PCO1 has the highest efficacy in suppressing inflammation and the subsequent effect on ROS generation and fatty liver change in the liver, which agrees well with the lowest serum AST and ALT levels. 

### 2.6. Kidney Tissue Analysis

Figure 6 presents the results of the H&E staining of the kidney. The ND group showed a compact, densely packed renal tissue structure with well-defined proximal and distal tubules. In contrast, the HCD feed group exhibited a loosely arranged and sparsely populated renal tissue structure with some lumen debris (indicated by the red arrow), highlighting the adverse impact of the HCD. The PCO-treated groups, precisely PCO1, exhibited a compact tissue framework with appropriately packed proximal and distal tubules without the lumen debris, implying the impact of PCO1 to mitigate HCD-induced kidney damage. Similarly, PCO2 and PCO3 also displayed a regular and compact proximal and distal tubule, but in some portions, a curved tubular-shaped structure that complies with new nephrons was also observed (as indicated by the green arrow in photos a1 and c1, Figure 6A) along with some lumen debris (indicated by the red arrow). The outcomes revealed the therapeutic impact of PCO1 against HCD-induced kidney damage.

The DHE fluorescent staining-based analysis revealed massive ROS generation in the HCD feed group, which was 2.6-fold higher (*p* < 0.001) than the ROS level quantified in the ND group (Figure 6). A co-treatment of HCD with PCO1 inhibited HCD-induced ROS production, as documented by a significant two-fold decrease (*p* < 0.001) in the DHE fluorescent intensity in the PCO1 group compared to that of the HCD group. In contrast, the supplementation of PCO2 and PCO3 did not influence HCD-induced ROS generation, suggesting the higher functionality of PCO1. Furthermore, the extent of apoptosis in each policosanol-supplemented group was evaluated using AO fluorescent staining. As shown in Figure 6, the least AO fluorescent intensity was observed in the ND alone group. In contrast, a 3.6-fold higher (*p* < 0.001) AO fluorescent intensity was observed in the HCD alone group, indicating increased apoptosis by hyperlipidemia. PCO1 and PCO3 reduced the HCD-induced apoptosis. A 2.7- (*p* < 0.001) and 1.7-fold (*p* < 0.01) lower AO fluorescent intensity was observed in the PCO1 and PCO2-treated groups compared to the HCD feed group. Compared to PCO3, a 1.6-fold lower AO fluorescent intensity was observed in the PCO1-treated group, signifying the higher efficacy of PCO1 to inhibit apoptosis. In contrast to PCO1, PCO2 did not prevent apoptosis induced by the HCD. These results collectively suggest the efficacy of PCO1 in preventing HCD-induced ROS generation, apoptotic cell death, and kidney damage.

### 2.7. Ovarian Tissue Analysis

After consumption, H&E staining (Figure 7A,B) showed that the HCD group had the highest previtellogenic oocyte content (~87%) and the lowest content of early and mature vitellogenic oocytes (~10% and 3%, respectively), compared to the ND group. On the other hand, the PCO1 group showed the highest content of mature vitellogenic oocytes (~23%), with the lowest content of premature oocytes (~63%). In contrast, the PCO2 and PCO3 groups showed a much higher content of premature oocytes (~74–84%) with a lower content of early and mature vitellogenic oocytes (~12–20%). 

AO staining showed that the HCD group exhibited the strongest green intensity, 5.6-fold higher than the ND group (Figure 7A,C), implying the association of high cholesterol consumption with increased cellular apoptosis. On the other hand, the PCO1 group showed a 72% lower green fluorescence intensity corresponding to the degree of apoptosis than the HCD alone group. In contrast, the PCO2 and PCO3 groups displayed a 36–44% lower AO-stained area than the HCD alone group, suggesting that Cuban policosanol exhibited the highest inhibition activity to suppress apoptosis in ovarian cells. Visualization of ROS by DHE staining showed that the HCD group had a 2.8-fold stronger red fluorescent intensity than the ND group, indicating higher ROS generation in oocytes in response to HCD supplementation (Figure 7A,C). In contrast, the PCO1 group showed the lowest red fluorescent intensity, which was 53% lower (*p* = 0.006) than that of the HCD group, highlighting the impact of PCO1 against HCD-induced ROS generation. In contrast, although the PCO2 and PCO3 groups showed a 33% and 29% lower DHE-stained area, respectively, than the HCD alone group, there was no significant decrease between the PCO2 and PCO3 groups. These results indicate that Cuban policosanol consumption resulted in the highest content of mature oocytes with the lowest extent of cellular apoptosis and ROS generation in the ovarian cells. In contrast, other policosanol had a weaker effect on protecting the ovarian tissue. 

### 2.8. Analysis of Testicular Cell 

H&E staining of the testis section showed that the ND group (photograph a1) had healthy seminiferous tubules with a full cell population adherent to the basal membranes without any notable gaps between the membranes and interstitium (Figure 8A), approximately 16% of the interstitial area (Figure 8B). In contrast, the HCD alone group (photograph b1) displayed irregularly outlined seminiferous tubules with disarranged cellular layers and a broken lamina basal membrane with the largest interstitial area of approximately 28%, indicating impaired spermatogenesis. The morphometric results of spermatogenesis in the HCD control group (photograph b1) showed that spermatids and sperm (mature forms) were smaller than those in the ND control group (photograph a1). On the other hand, the PCO1 group showed an increase in the area of spermatids, as shown in Figure 8A (photograph c1), with the smallest interstitial area around 18% among HCD groups, which was approximately 36% smaller than the HCD alone group, respectively (*p* < 0.001). In contrast, the PCO2 and PCO3 groups showed 26% and 28% of the average interstitial areas, respectively, similar to that of the average interstitial areas of the HCD-alone group (Figure 8B).

AO staining and DHE staining showed that the HCD group had a 4.6-fold and 3.8-fold higher green and red intensity, respectively, than the ND group (Figure 8C), suggesting that high cholesterol consumption caused a remarkable increase in cellular apoptosis and ROS generation in the testicular cell. On the other hand, the PCO1 group showed 78% and 83% lower green and red intensity, respectively, than the HCD alone group, indicating the highest cytoprotection activity. In contrast, the PCO2 and PCO3 groups had a non-significant effect on inhibiting HCD-induced apoptosis and ROS generation. The combined histology results of the testicular cells suggested that the Cuban policosanol had the highest protective activity with the lowest extent of cellular apoptosis and ROS generation in spermatogenesis.

### 2.9. Embryo Production Ability and Size of the Ovary and Testis

At week 8, fertility-based egg production per mating was remarkably lower in the HCD group (~49 ± 2 eggs/mating) than the ND group (~150 ± 41 eggs/mating), suggesting a 78% decrease in egg laying ability by HCD consumption, as shown in Figure 9A. In contrast, the PCO1 group showed 244 ± 23 eggs/mating, presenting a 5.0-fold (*p* = 0.015) and 1.3-fold (*p* = 0.032) higher egg production ability than the HCD and ND groups. In contrast, the PCO2 and PCO3 groups showed 59 ± 2 and 86 ± 13 eggs/mating, respectively, similar to the HCD group. 

Stereoimage observations showed that the PCO1 group had the largest ovary and testis, while the PCO3 group and HCD alone group had the smallest (Figure 9B). Measurements of the ovary wet weight showed that the HCD group showed approximately 30 mg of ovary wet weight, which was 6% higher than the ND group (~27 mg), but the difference was not significant. On the other hand, the PCO1 group showed the highest weight (~35 mg), while the PCO2 and PCO3 groups showed the lowest (~22–23 mg) (Figure 9C). Interestingly, the HCD alone group showed the lowest wet weight of the testis of ~2.0 mg, 20% smaller testis weight than the ND group (~2.5 mg), suggesting that the HCD consumption caused more impairment in the testis rather than the ovary. On the other hand, the PCO1 group had the largest testis weight (~3.2 mg), which was 1.6-fold higher than that of the HCD group. On the other hand, the PCO2 and PCO3 groups showed smaller testis weights than the PCO1 group; nevertheless, they were 1.4 and 1.5-fold higher than the HCD alone group. These results suggest that HCD consumption impaired the embryo production ability through degeneration of the reproduction organs, particularly the dwindling of the testis. On the other hand, the degeneration of fertility could be restored using Cuban policosanol consumption by strengthening the reproductive organs, as shown in Figure 9. 

### 2.10. Embryo Survivability and Developmental Morphology

Embryos from the ND and HCD groups exhibited 92 and 53% survivability during 120 h pro-fertilization, suggesting that the HCD group exhibited remarkably lower survivability, 43% lower than the ND group, as shown in Figure 10A. Embryos from the HCD group showed remarkably slower developmental speed and deformity with attenuated eye pigmentation and tail elongation (Figure 10B). On the contrary, embryos from the PCO1 group showed the highest survivability of approximately 98% (*p* = 0.003 vs. HCD alone) and the fastest developmental speed. In contrast, the PCO2 and PCO3 groups showed lower survivability (~68% and 46%, respectively), with a slower developmental speed than the PCO1 group (Figure 10).

At 120 h post-fertilization (Figure 11), the larvae in the ND group (Appendix A) showed a much faster and distinct swimming pattern with longer distance movement than those in the HCD group, as shown in Appendix A. This suggests that HCD consumption impaired embryo development, causing a weaker swimming pattern and attenuation of developmental speed with morphological defects, as shown in the zoomed image of zebrafish. On the other hand, as shown in Figure 11, the PCO1 group (Appendix A) showed a much faster swimming pattern with longer distance movement than those of the HCD alone group, with a developmental speed similar to that of the ND group. In contrast, the PCO2 (Appendix A) and PCO3 (Appendix A) groups showed a much weaker and slower swimming pattern similar to the HCD alone group with an attenuation of the developmental speed. These results strongly suggest that the co-consumption of Cuban policosanol induced a faster developmental speed and higher survivability by enhancing the reproduction organs impaired by HCD consumption. On the other hand, the co-consumption of Chinese or American policosanol did not improve the impairment of reproduction organs and the developmental defects under HCD.

## 3. Discussion

Several policosanol brands, at least 25 products, are currently selling on the global market, with a 5–40 mg daily dosage, from many different natural sources and countries of origin, such as Cuba, China, and the USA. Sales of policosanol worldwide have increased rapidly because of its beneficial efficacy in treating dyslipidemia and hypertension and the high demand for natural-based therapies to avoid synthetic and chemical additives [11,31]. Despite the growing market size, there has been insufficient information on the efficacy and side effects of the products with many brand names, which can confuse consumers. Moreover, there has been almost no study comparing policosanols from different plant sources, such as sugarcane, rice bran, oat, and wheat, despite their reported beneficial effects [32]. This study compared the in vivo efficacy of various policosanols from rice bran and sugarcane wax produced in different countries. Although this study compared the in vivo efficacy of policosanols from sugarcane wax and rice bran, which was a total amount of 980 mg on the label in the preceding study [20], no study has compared sugarcane and rice bran, which had a total amount of 400 mg on the label. Because many similar policosanol products from rice bran and sugar cane with different compositions of aliphatic alcohols might have different physiological activities, it is necessary to compare the in vitro and in vivo evaluations via rHDL synthesis and supplementation with HCD, respectively. 

In the current study, HCD supplementation for twelve weeks resulted in the lowest survivability (Figure 1A) and the highest increase in body weight (Figure 1B) with severe dyslipidemia (Figure 2). The impairment of the hepatic function in the HCD group (Figure 3) was associated with the elevation of hepatic inflammation and fatty liver changes (Figure 4 and Figure 5). The HCD group showed the most kidney, ovary, and testis damage in morphology and ROS production with cellular apoptosis (Figure 6, Figure 7 and Figure 8). The organ damage was linked to the loss of reproduction function (Figure 9) with developmental defects in the embryos (Figure 10). The lowest egg-laying ability was associated with the most attenuation of developmental speed and retardation of swimming ability in the HCD group (Figure 11 and Appendix A). These results suggest that hyperlipidemia could provoke severe infertility and embryo defects via damage to major organs for reproduction, such as the ovary and testis (Figure 7, Figure 8 and Figure 9). These findings concur with previous reports, implying that hyperlipidemia is intimately associated with female infertility [33] and male infertility [34] via obesity-related inflammation and oxidative stress [35,36]. In particular, low HDL-C is associated more specifically with an impairment of spermatogenesis and erectile dysfunction [37,38]. Cholesterol is essential in the male reproduction system via spermatogenesis, steroidogenesis, and testosterone production. Sertoli cells, which promote sperm production, block the passage of LDL at the blood-testis barrier (BTB) but allow the entry of HDL to the seminiferous tubules [39]. Cholesterol from HDL is essential for spermatogenesis because it serves as a fuel for Sertoli cells. In the same context, boys have a strong demand for cholesterol from HDL for spermatogenesis during the pubertal period, resulting in a lower HDL-C in adulthood [40]. Erectile dysfunction is frequently associated with the incidence of cardiovascular disease through low HDL-C [41]. Overall, the induction of dyslipidemia, particularly low serum HDL-C, also impaired the liver and kidney, which are associated with the exacerbation of infertility and multi-organ dysfunction via severe systemic inflammation, such as sepsis [42]. 

On the other hand, among the three policosanols, co-supplementation of Cuban policosanol produced the highest survivability and lowest body weight gain (Figure 1) with the best improvement in the lipid profile (Figure 2), hepatic functions (Figure 3), and histological morphology of the major organ (Figure 4, Figure 5, Figure 6, Figure 7 and Figure 8). In particular, the Cuban policosanol group showed the lowest AST and ALT levels with the lowest fatty liver change, IL-6 production, and ROS production (Figure 3, Figure 4 and Figure 5). These improvements in liver function concurred with a previous animal model study for eight weeks of Cuban policosanol consumption, using zebrafish and spontaneous hypertensive rats [43,44], which showed a remarkable decrease in neutrophil infiltration and ROS production in hepatocytes. Moreover, twelve weeks of consumption in a randomized human study with Japanese subjects also showed a significant decrease in the serum AST, ALT, ALP, γ-GTP, blood urea nitrogen, uric acid, and glycated hemoglobin levels [45,46]. These clinical results align with the current results that showed the protective effect of Cuban policosanol on the liver and kidney.

Co-supplementation of Cuban policosanol also resulted in the largest size and weight of the ovary and testis with the highest egg-laying ability and the fastest embryo development. In contrast, the Chinese and American policosanol groups showed much smaller ovary and testis sizes and low fertility, similar to the HCD alone group.

These results suggest that the increase in HDL-C and HDL functionality enhancement by Cuban policosanol contributes to lower inflammation and protection of the liver, kidneys, ovary, and testis [15,20]. Regarding efficacy in treating dyslipidemia and multi-organ dysfunction, there were distinct differences between Cuban policosanol and the other policosanols (Chinese policosanols, such as Xi’an Natural, Xi’an Realin, Shaanxi, and American policosanol). Furthermore, Cuban policosanol supplementation remarkably augmented the reproduction ability and developmental speed/morphology. To the best of the authors’ knowledge, this study is the first to show that treating dyslipidemia and inflammation could help recover reproduction ability and promote development via the protection of major organs, such as the liver, ovary, and testis. Overall, Cuban policosanol consumption enhanced fertility and ameliorated the ovarian and testicular functions to produce healthy embryo development beyond improving hepatic function, dyslipidemia, and inflammation. The study’s outcome signifies the relative impact of three distinct policosanol brands derived from diverse sources and countries in mitigating various parameters affected by HCD consumption in zebrafish. Further investigation into the bioavailability or metabolism of policosanols from various sources is essential to uncover and understand how these factors impact the bioactivity of policosanols.

## 4. Materials and Methods

### 4.1. Materials

Cuban sugarcane wax alcohol (Raydel^®^), PCO1, was procured from the National Center for Scientific Research (CNIC), Havana, Cuba, via Raydel Australia (Thornleigh, Sydney, Australia). Rice bran extracted Chinese policosanol (PCO2, total wax alcohol 400 mg/g on the label) was obtained from Shaanxi Pioneer Biotech (Xi’an, China). American policosanol was purchased from Garuda International (Lesstanol^®^, Exter, CA, USA). All raw materials of each policosanol were analyzed for the ingredient compositions of aliphatic alcohols using the gas chromatography described elsewhere [15,16]. 

### 4.2. Maintenace of Zebrafish and Policosanol Supplementation

Zebrafish and embryos were managed by the designated protocols outlined in the Guide for the Care and Use of Laboratory Animals [47]. The Animal Care and Use Committee at Raydel Research Institute (approval code RRI-20-003, approval date 3 January 2020, Daegu, Republic of Korea) authorized the zebrafish procedures in adherence to the ARRIVE (Animal Research: Reporting of In Vivo Experiments) guidelines 2.0 [48]. The zebrafish were accommodated in a controlled environment at an invariant temperature of 28 °C throughout the investigation, following a 12 h alternative cycle of light and dark. Zebrafish were fed a diet comprising regular tetrabits (Tetrabit D49304, Melle, Germany).

The zebrafish (aged 10 weeks) were arbitrarily assigned to five cohorts (refer to Table 1), with each cohort consisting of 70 zebrafish (*n* = 70). Over twelve weeks, the zebrafish were incorporated into a Tetrabit diet (Gmhb D49304, Melle, Germany) enriched with one of the various policosanol variants (final concentration 0.1%, *w*/*w*) and a high cholesterol diet (HCD, 4% cholesterol, *w*/*w*). The groups were categorized as follows: a control group exposed to a normal diet (ND), a group exposed to a 4% high-cholesterol diet (HCD), PCO1 (exposed to Cuban policosanol, Raydel^®^ + HCD), and PCO2 (exposed to Chinese policosanol, Shaanxi + HCD), and PCO3 (exposed to American policosanol, Garuda + HCD). Before introducing each variant of policosanol, all groups, except for the ND alone group, experienced a four-week acclimation period to the HCD. Following the acclimation period, the zebrafish were provided with a daily policosanol supplement (0.01 mg) in Tetrabit (10 mg) at specific intervals, both in the morning (9 am) and evening (6 pm), resulting in a total policosanol intake of 0.02 mg for each zebrafish. 

### 4.3. Blood Collection and Analysis

Blood samples were obtained from various zebrafish groups following twelve weeks of supplementation to analyze plasma lipid levels, employing the procedure defined elsewhere [20]. The levels of total cholesterol (TC) and triglycerides (TG) were assessed using a diagnostic kit (Cholesterol, T-CHO, and TG, Cleantech TS-S; Wako Pure Chemical, Osaka, Japan). High-density lipoprotein cholesterol (HDL-C) (AM-202), alanine transaminase (ALT) (AM-103K), and aspartate transaminase (AST) (AM-201) were assessed utilizing a diagnostic kit (Asan Pharmaceutical, Hwasung, Korea). The Appendix A includes a comprehensive description of the detection procedure (Appendix A). 

### 4.4. Examination of Liver Tissue 

Zebrafish livers from different groups were surgically removed after sacrifice. Liver tissue sections, each 7μm thick, were subjected to processing for Hematoxylin and Eosin (H&E), oil red O (ORO), and dihydroethidium (DHE) fluorescent staining [49]. Hepatic tissue IL-6 production was quantified through immunohistochemical staining using a previously described method [50]. The hepatic tissue section (7 μm thick) was immersed with IL-6 specific primary antibody (200 times diluted; (ab9324, Abcam, London, UK) and incubated at 4 °C for 18 h. After that, the tissue section was developed using horseradish peroxidase (HRP)-conjugated anti-IL-6 antibody (1000 time diluted) using the EnVison + System HRP labeled polymer kit (Code K4001, Dako, Glostrup, Denmark). The Appendix A provided a detailed account of the histological analysis, encompassing the procedures for fluorescent and immunohistochemical staining.

### 4.5. Histological Examination of Kidney

The zebrafish were sacrificed by hypothermic shock, and the kidney was immediately recovered surgically. The tissue was preserved in 10% formalin, followed by embedding in paraffin and fixing at low temperature for the histological examination. Finally, a tissue section (7 μm thick) was prepared using a cryomicrotome (Leica, Nussloch, Germany), and the section was processed for hematoxylin and eosin staining using a previously described method [20]. The extent of ROS production and apoptosis in the tissue section was examined by DHE and AO fluorescent staining, respectively, using a previously described method [49,51]. Briefly, the DHE (final 30 μM) and AO (final 5 μg/mL) stains were spread over the tissue section and allowed to react in the dark for 30 min. Subsequently, the tissue section was rinsed and envisioned under a fluorescent microscope (Nikon Eclipse TE2000, Tokyo, Japan) at excitation (Ex) and emission (Em) wavelengths of 585 nm (Ex) and 615 nm (Em) for DHE and 505 nm (Ex) and 535 nm (Em) for the AO stained area.

### 4.6. Analysis of Ovarian Tissue 

Ovarian samples from distinct groups were surgically extracted post-sacrifice. The ovarian tissue section (7 μm thick) underwent Hematoxylin and Eosin (H&E) staining to differentiate oocytes developmental stages, i.e., pre-vitellogenic, early-vitellogenic, and mature-vitellogenic stages of the ovary, as outlined earlier [52,53]. Dihydroethidium (DHE) fluorescent staining and acridine orange (AO) fluorescent staining were conducted to assess ROS production and apoptosis extent, respectively, following established protocols [49,51]. For a comprehensive description of the histological analysis procedure, including fluorescent staining, refer to the Appendix A.

### 4.7. Analysis of Testis Tissue 

Testicular samples from the separate groups were surgically obtained after sacrifice. The testis tissue section, 7 μm thickness, underwent processing for H&E staining as described previously [54,55]. Quantifying reactive oxygen species (ROS) production and apoptosis extent was conducted using DHE fluorescent staining and AO fluorescent staining, respectively, following the established procedures [49,51]. A comprehensive histological analysis, including details on the fluorescent staining process, is mentioned in the Appendix A.

### 4.8. Mating and Embryo Production

During supplementation of the designated diet, zebrafish embryos were produced at week 8 and counted in each group. Zebrafish of both genders within a common group were chosen, combined at a ratio of 1:2, and separated in the spawning tank by a physical partition. Following 16 h of isolation, the dividers in the spawning tank were eliminated, permitting uninterrupted mating for about 1 h. Finally, the produced embryos were collected and counted to compare the embryo numbers in each group. In each group, the collected embryos were washed with egg water and incubated for 120 h at 28 °C to monitor the survivability, developmental speed, and developmental defects [56,57]. 

### 4.9. Statistical Evaluation

The outcomes are expressed as mean ± SEM for each set of experiments performed in triplicates. Group comparisons were performed through a one-way ANOVA using SPSS software (version 29.0, SPSS, Inc., Chicago, IL, USA). Subsequently, Dunnett’s post hoc analysis was utilized to identify significant differences among the groups, with a significance level set at *p* < 0.05. 

## 5. Conclusions

Cuban policosanol consumption (0.1% in the diet, *w*/*w*) for twelve weeks demonstrated potent efficacy in treating dyslipidemia and inflammation. It resulted in the highest survivability with advantageous effects to protect the liver, kidney, ovary, and testis from the comparison study with Cuban, Chinese, and American policosanol. At week 8, the Cuban policosanol group showed the highest embryo reproduction ability with the highest survivability of larvae, the fastest developmental speed, and normal developmental morphology. These results suggest that Cuban policosanol supplementation could rescue the multi-organ dysfunction and infertility caused by HCD consumption, with the highest survivability in adult zebrafish and their offspring. Nevertheless, conducting a subsequent study involving extended policosanol supplementation under HCD conditions is imperative to validate this assertion. 

## Figures and Tables

**Figure 1 pharmaceuticals-17-00066-f001:**
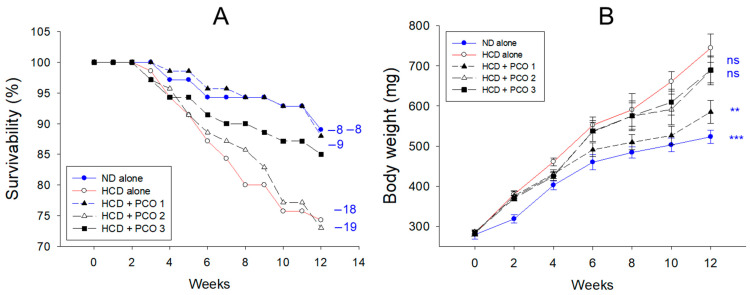
Survivability (**A**) and body weight (**B**) changes were assessed over twelve weeks of policosanol supplementation under a high-cholesterol diet. Blue font numerals indicate zebrafish mortality during the twelve weeks in each group. Data are presented as mean ± SEM, and group differences were statistically evaluated using a one-way analysis of variance (ANOVA) with Dunnett’s post-hoc test. ** and *** refers to the significant difference at *p* < 0.01 and *p* < 0.001, respectively, compared HCD control; ns, not significant compared HCD alone. HCD refers to a high cholesterol diet, ND to a normal diet, PCO1 to Raydel policosanol, PCO2 to Shaanxi policosanol, and PCO3 to Garuda policosanol.

**Figure 2 pharmaceuticals-17-00066-f002:**
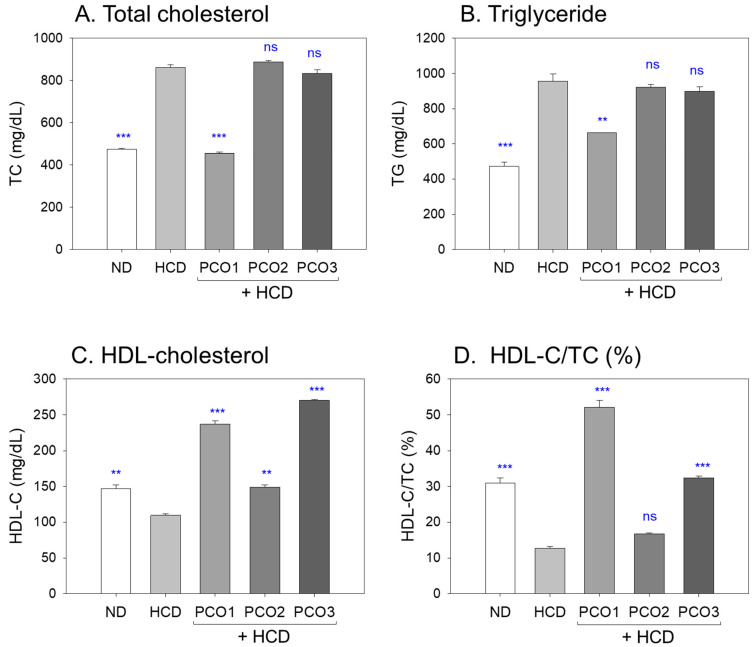
Measurement of total cholesterol (TC), triglyceride (TG), high-density lipoprotein cholesterol (HDL-C), and the ratio of HDL-C/TC (%) in the blood following twelve weeks of policosanol (PCO) supplementation amidst a high cholesterol diet. Data are presented as mean ± SEM, and group differences were statistically evaluated using a one-way analysis of variance (ANOVA) with Dunnett’s post-hoc test. ** and *** refers to the significant difference at *p* < 0.01 and *p* < 0.001, respectively, compared HCD control; ns, not significant compared the HCD control. HCD refers to a high cholesterol diet, ND to a normal diet, PCO1 to Raydel policosanol, PCO2 to Shaanxi policosanol, and PCO3 to Garuda policosanol.

**Figure 3 pharmaceuticals-17-00066-f003:**
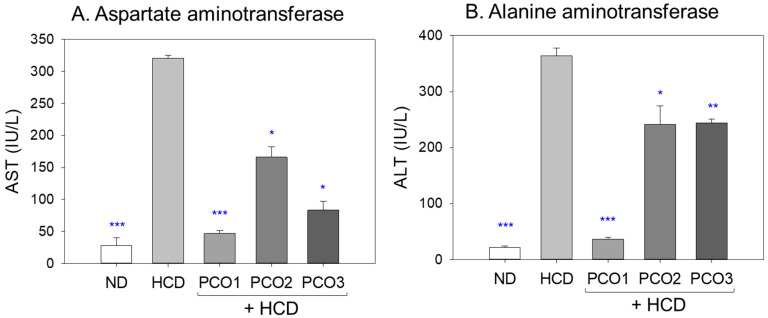
Evaluating liver function biomarkers, (**A**) aspartate aminotransferase (AST) and (**B**) alanine aminotransferase (ALT) following twelve weeks supplementation of policosanol in the presence of high cholesterol diet (HCD). The data is presented as mean ± SEM. Statistical significance among the groups was assessed using a one-way analysis of variance (ANOVA) with Dunnett’s post-hoc test, comparing each group to the HCD group. HCD refers to a high cholesterol diet, ND to a normal diet, PCO1 to Raydel policosanol, PCO2 to Shaanxi policosanol, and PCO3 to Garuda policosanol. *, *p* < 0.05 compared to the HCD control; **, *p* < 0.01 compared to the HCD control; ***, *p* < 0.001 compared to the HCD control.

**Figure 4 pharmaceuticals-17-00066-f004:**
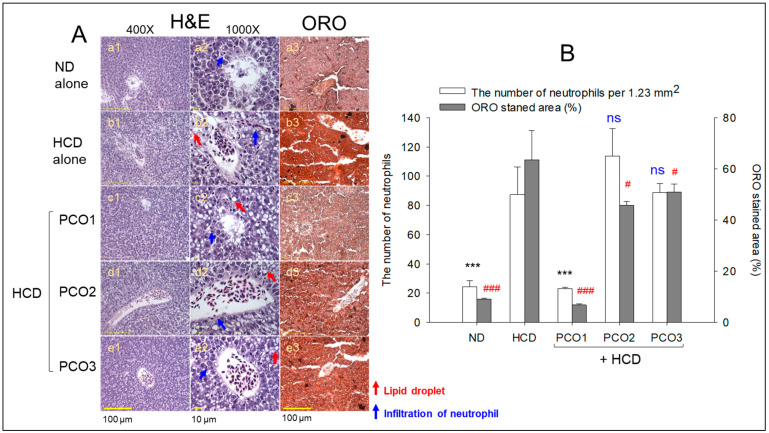
Hepatic morphology and fatty liver change after twelve weeks of supplementation of each variant of policosanol (PCO) in the presence of high cholesterol diet (HCD) consumption. (**A**) Hematoxylin & Eosin (H&E) staining revealed the infiltration of neutrophils (indicated by blue arrows) and the presence of lipid droplets (indicated by red arrows). Images (a1–e1) and (a2–e2) were captured at magnification of 400× and 1000×, respectively. Oil-red O staining at 400× magnification (images a3–e3). [scale bar: 100 μm]. (**B**) Quantification analysis using Image J software (http://rsb.info.nih.gov/ij/ accessed on 17 May 2023) involved counting the number of infiltered neutrophils from H&E staining and assessing the oil-red O-stained area. The quantification was performed in designated areas (1.23 mm^2^) across five views, providing a semi-quantitative estimation of neutrophil infiltration. The data is presented as mean ± SEM. Statistical significance among the groups was assessed using a one-way analysis of variance (ANOVA) with Dunnett’s post-hoc test. *** denote *p* < 0.001, compared to the HCD alone group for neutrophil infiltration. At the same time, ^#^ and ^###^ indicate *p* < 0.05 and *p* < 0.001, respectively, compared to the HCD alone group for the Oil-red O-stained area. ns, not significant. HCD refers to a high cholesterol diet, ND to a normal diet, PCO1 to Raydel policosanol, PCO2 to Shaanxi policosanol, and PCO3 to Garuda policosanol.

**Figure 5 pharmaceuticals-17-00066-f005:**
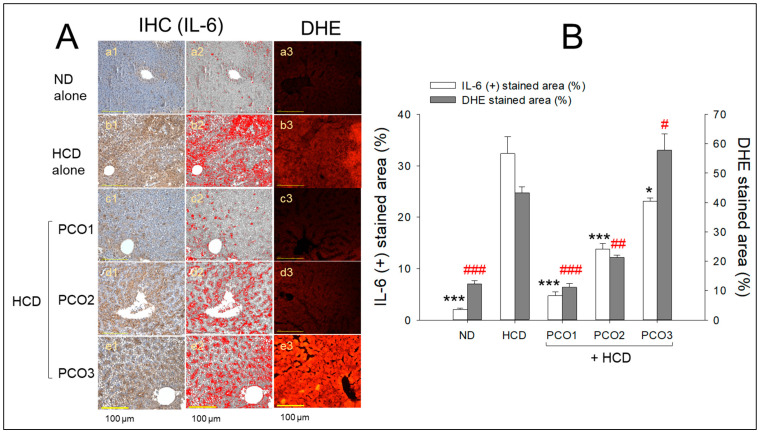
Comparative effect of twelve-week supplementation of various policosanol (PCO) types under a high cholesterol diet (HCD) on the production of interleukin (IL)-6 and generation of reactive oxygen species (ROS) in the hepatic tissue of zebrafish. (**A**) Illustrated images (a1–e1) depict IL-6-stained hepatic tissue determined using immunohistochemistry (IHC). The brown coloration of the IL-6 stained area interchanged with red color (a2–e2) to enhance the envision of the IL-6 stained area using Image J software (http://rsb.info.nih.gov/ij/ accessed on 16 June 2023). The ROS levels were assessed through the dihydroethidium (DHE)-fluorescent sustained area (a3–e3) [scale bar: 100 μm]. (**B**) Quantifying IL-6 and DHE stained areas using Image J software. The data are presented as mean ± SEM. Statistical differences of multiple groups were compared using a one-way analysis of variance (ANOVA) with Dunnett’s post-hoc test. * and *** signify *p* < 0.05 and *p* < 0.001, respectively, compared to the HCD control for the IL-6-stained area. Concurrently, ^#^, ^##^, and ^###^ indicate *p* < 0.05, *p* < 0.01, and *p* < 0.001, respectively, versus the HCD control for the DHE stained area. HCD refers to a high cholesterol diet, ND to a normal diet, PCO1 to Cuban policosanol, PCO2 to Chinese policosanol, and PCO3 to American policosanol.

**Figure 6 pharmaceuticals-17-00066-f006:**
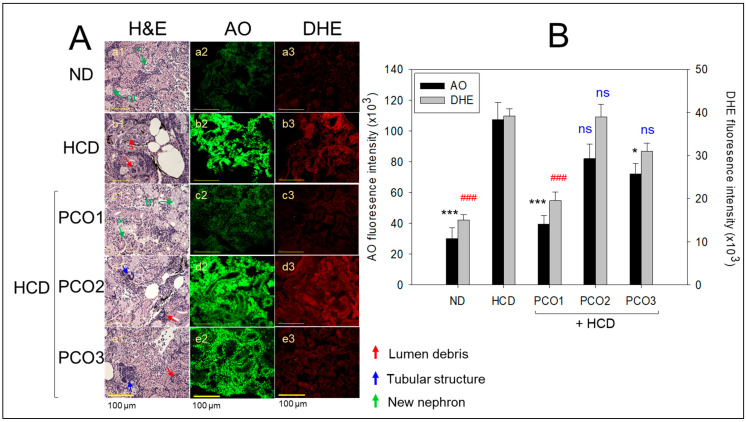
Comparative effect of policosanol supplementation for twelve weeks under a high-cholesterol diet (HCD) consumption on kidney morphology, reactive oxygen species (ROS) generation, and apoptosis extent in adult zebrafish. (**A**) Hematoxylin and eosin (H&E), acridine orange (AO), and dihydroethidium (DHE) staining were used to evaluate the morphological changes, the extent of apoptosis, and ROS generation, respectively, in the kidney. PT and DT represent the proximal and distal tubules, while the red, blue, and green arrows represent luminal debris, tubular structures, and new nephrons, respectively. [Scale bar: 100 μm]. (**B**) Image J software was intended to assess the fluorescent strength concerning AO and DHE-stained areas. HCD, high-cholesterol diet; ND, normal diet; PCO1, Cuban policosanol; PCO2, Chinese policosanol; PCO3, American policosanol. Data are shown as mean ± SEM. * and *** *p* < 0.05 and *p* < 0.001, respectively, versus the HCD alone group for AO fluorescent intensity. ^###^, *p* < 0.001 compared to the HCD alone group for the DHE fluorescent intensity, ns, not significant.

**Figure 7 pharmaceuticals-17-00066-f007:**
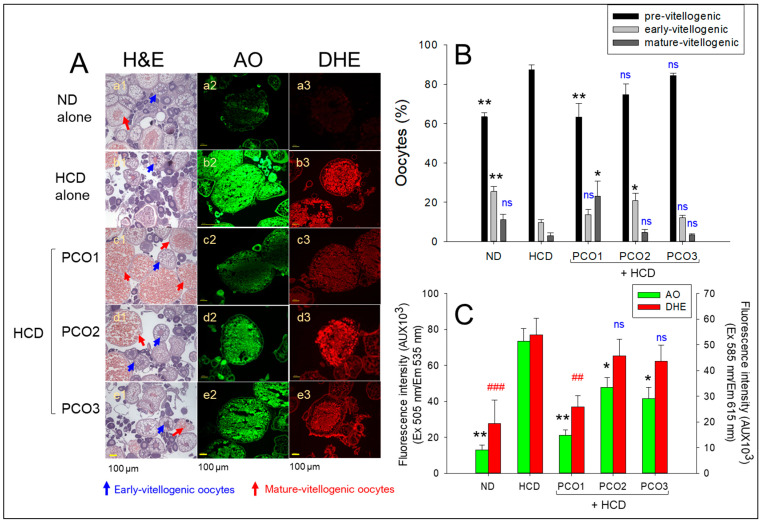
Comparative effect of policosanol supplementation for twelve weeks under a high cholesterol diet (HCD) consumption on ovarian cell morphology, reactive oxygen species (ROS) generation, and apoptosis extent in adult zebrafish. (**A**) Hematoxylin and eosin (H&E), acridine orange (AO), and dihydroethidium (DHE) staining were used to evaluate the morphological changes, the extent of apoptosis, and ROS production, respectively, in the ovarian tissue. Blue and red arrows indicate early and mature vitellogenic oocytes, respectively. [Scale bar: 100 μm]. (**B**) Percentage quantification of different stages of oocytes. * and ** *p* < 0.05 and *p* < 0.01, respectively, versus the HCD alone group. (**C**) Image J software was intended to assess the fluorescent strength concerning AO and DHE-stained areas. Data in the bar graphs are represented as mean ± SEM. * and ** *p* < 0.05 and *p* < 0.01, respectively, versus the HCD alone group for AO fluorescent intensity. ^##^ and ^###^ *p* < 0.01 and *p* < 0.001, respectively, compared to the HCD alone group for the DHE fluorescent intensity, ns, not significant. HCD, high cholesterol diet; ND, normal diet; PCO1, Cuban policosanol; PCO2, Chinese policosanol; PCO3, American policosanol.

**Figure 8 pharmaceuticals-17-00066-f008:**
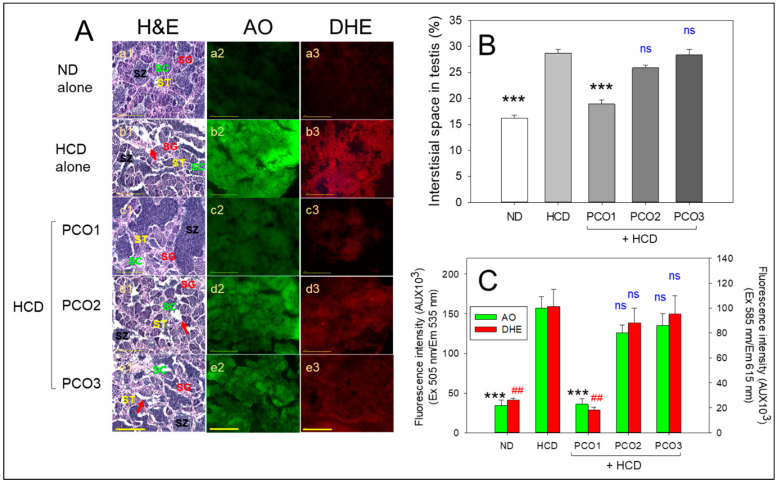
Investigation of the comparative effect of policosanol supplementation over twelve weeks in the presence of high cholesterol diet (HCD) on the morphology of testicular cells, generation of reactive oxygen species (ROS), and the extent of the apoptosis in adult zebrafish. (**A**) Morphological changes, apoptosis extent, and ROS production in the testis were assessed using Hematoxylin and eosin (H&E), acridine orange (AO), and dihydroethidium (DHE) staining, respectively. Various cell types were identified: SG (spermatogonia), SZ (spermatozoa), ST (spermatid), and SC (spermatocytes). The red arrow highlights interstitial spaces in the seminiferous tubules’ void area. [Scale bar: 100 μm]. (**B**) Quantifying interstitial spaces in seminiferous tubules of the testis employing Image J software. *** indicates *p* < 0.001 versus the HCD alone group. (**C**) Image J software was employed to evaluate fluorescent strength in AO and DHE-stained areas. Data in the bar graphs are represented as mean ± SEM. HCD, high cholesterol diet; ND, normal diet; PCO1, Cuban policosanol; PCO2, Chinese policosanol; PCO3, American policosanol. *** indicate *p* < 0.001 versus HCD alone group for AO fluorescent intensity. *^##^* shows *p* < 0.01 versus HCD alone group for DHE fluorescent intensity, ns, not significant.

**Figure 9 pharmaceuticals-17-00066-f009:**
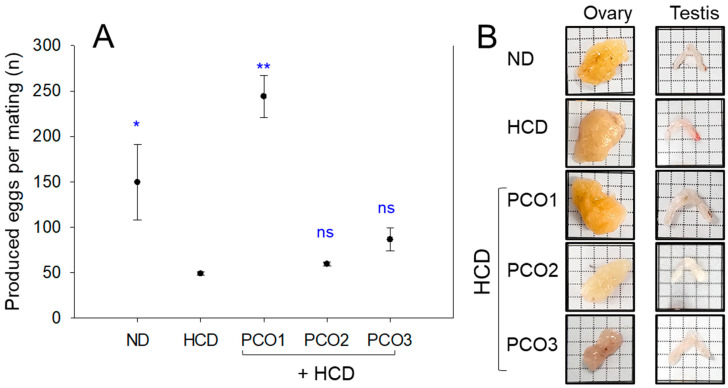
Embryos production ability and wet weight of reproductive organs in each group. HCD, high cholesterol diet; ND, normal diet; PCO1, Cuban policosanol; PCO2, Chinese policosanol; PCO3, American policosanol. (**A**) Embryo production number per mating at week 8 under HCD consumption, along with different policosanols. * and ** signify *p* < 0.05 and *p* < 0.01 versus HCD alone group. ns, not significant. (**B**) Comparison of the organ size and wet weight in the ovary and testis at week 12 of policosanol under HCD consumption. All observations were made at 10× magnification. Comparison of the ovary weight (**C**) and testis weight (**D**) among each group at week 12 of policosanol supplementation under HCD consumption. Data are shown as mean ± SEM. *, *p* < 0.05 versus HCD alone group.

**Figure 10 pharmaceuticals-17-00066-f010:**
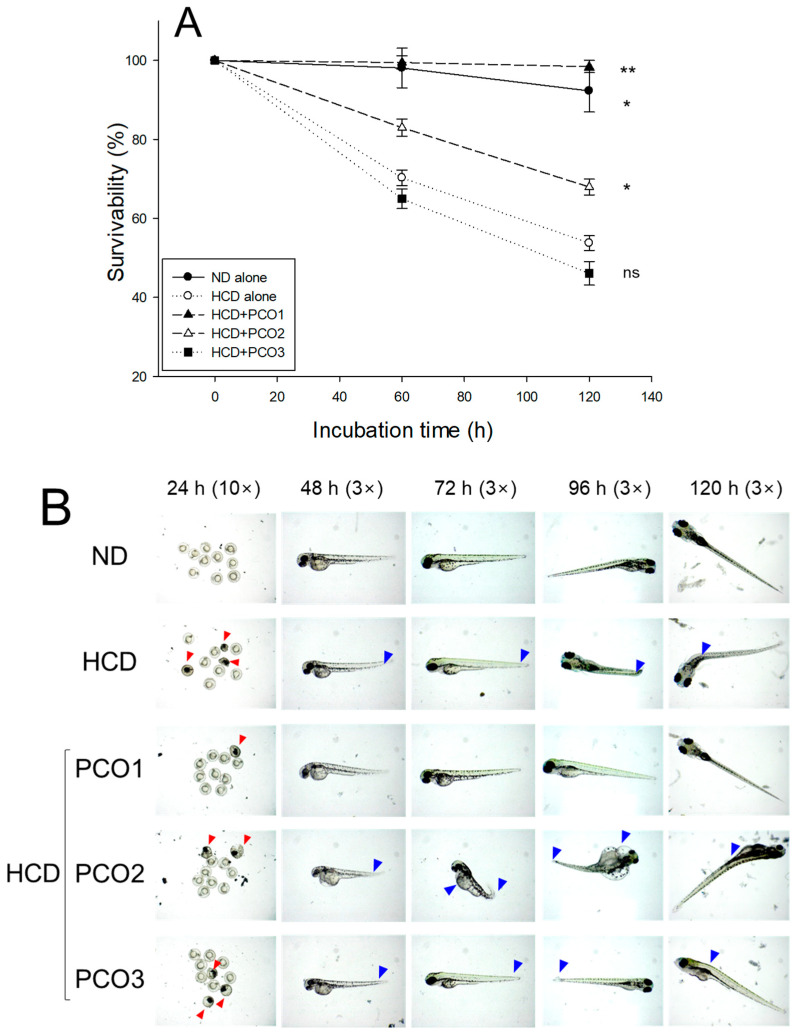
Embryo survivability and developmental morphology of produced eggs in each group from mating at week 8. HCD, high cholesterol diet; ND, normal diet; PCO1, Cuban policosanol; PCO2, Chinese policosanol; PCO3, American policosanol. (**A**) Embryonic survivability during 120 h post-fertilization. Data are shown as mean ± SEM. * and ** signify *p* < 0.05 and *p* < 0.01, respectively, compared to the HCD control. ns, not significant. (**B**) Comparison of the morphological changes and developmental speed. The red arrowhead indicates embryo death; the blue arrowhead indicates a deformity with an attenuation of eye pigmentation and tail elongation.

**Figure 11 pharmaceuticals-17-00066-f011:**
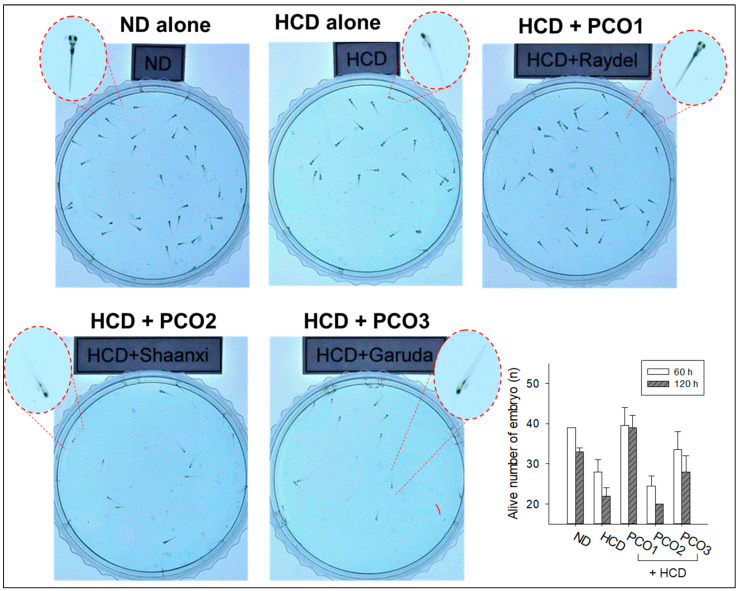
Still image of zebrafish larvae at 120 h post-fertilization (magnification 2×). The embryos were produced by mating at week 8 during HCD and policosanol supplementation. Inset graph showed the live number of zebrafish embryos at 60 h and 120 h post fertilization. Data are shown as mean ± SEM. HCD, high cholesterol diet; ND, normal diet; PCO1, Cuban policosanol; PCO2, Chinese policosanol; PCO3, American policosanol. The inset graph shows the live larvae numbers at 60 and 120 h. Each video file is available in the Appendix A.

**Table 1 pharmaceuticals-17-00066-t001:** Various combinations of policosanol combined with a high-cholesterol diet (HCD) over twelve weeks.

		ND			HCD	
	Cohorts (*n* = 70 in each group)	Control (*n* = 70)	Control (*n* = 70)	PCO1 Cuban CNIC Sugarcane Raydel^®^ (*n* = 70)	PCO2 Chinese Shaanxi Rice bran 400 (*n* = 70)	PCO3 American Garuda Sugarcane Lesstanol^®^ (*n* = 70)
Diet composition (%)	Tetrabits ^1^	100	96	95.9	95.9	95.9
	Cholesterol (%, *w*/*w*)	-	4	4	4	4
	PCO (%, *w*/*w*)	-	-	0.1	0.1	0.1
	Octacosanol content within in PCO (mg/g) ^2^	-	-	692 (70.5%) ^3^	56 (7.6%)	546 (60.7%)
	Final total PCO amount (mg) ^2^			982	739	900
	Total PCO amount on label ^4^			982	400	900
Body weight at week 0		279 ± 11	282 ± 5	284 ± 7	285 ± 6	283 ± 8

^1^ Tetrabits, a brand name of the zebrafish diet, were purchased from TetrabitGmbh (Melle, Germany). 47.5% crude protein, 6.5% crude fat, 2.0% crude fiber, 10.5% crude ash, containing vitamin A (29,770 IU/kg), vitamin D3 (1860 IU/kg), vitamin E (200 mg/kg), and vitamin C (137 mg/kg). ^2^ Derived from Reference [15]. ^3^ Percentage of octacosanol content in the total amount of policosanol. ^4^ Derived from Reference [16]. ND, normal diet; HCD, high cholesterol diet; PCO, policosanol.

## Data Availability

The data used to support the findings of this study are available from the corresponding author upon reasonable request.

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
