# Peer review of "Consumption of Policosanol (Raydel®) Improves Hepatic, Renal, and Reproductive Functions in Zebrafish: In Vivo Comparison Study among Cuban, Chinese, and American Policosanol"

_pharmaceuticals, 2023, doi:10.3390/ph17010066_

Round 1

Reviewer 1 Report

Comments and Suggestions for Authors

The authors Kyung-Hyun Cho et al. present a very carefully written manuscript “Consumption of Cuban policosanol (Raydel®) improves functions of liver, kidney, ovary, and testis in hyperlipidemic zebrafish to exhibit the highest reproduction ability and survivability with the lowest inflammation: In vivo comparison study among Cuban, Chinese, and American policosanol” based on obtaining a large number of reliable results. A manuscript of this level will certainly be of interest to the readers of the journal Pharmaceuticals and can be published after correcting minor comments related mainly to the design of the manuscript.

1. I would like to recommend the authors to change the title of the manuscript, presenting it in a more concise form. Currently, it is too large and detailed, which does not meet the requirements for the titles of articles.

2. The authors present a very extensive and detailed introduction with relevant references to literary sources. I really want to thank the authors for using up-to-date information! I would like to recommend the authors to supplement the introduction (at the end of this section) with a summary of the above with information about what results will be described in the further text of the manuscript.

3. I would really like to recommend the authors of the manuscript to change the way the results are presented for “HCD alone" in Figure 1. It is currently extremely poorly readable.

4.The authors should correct the abbreviation "alanine aminotransferase" on line 357.

5.The authors should carefully double-check all the captions to the figures and the necessary designations, in particular:

- The authors should add a transcript of the characters # and ### in the caption to Figure 4, since this is not currently obvious.

- In the caption to Figure 4A, the authors refer to the blue and red arrows, but they are completely invisible in the figure. There are also no symbols in the figures, except for "a".

- On line 449, the authors refer to the green arrow in Figure 6, but there are no arrows with this color.

- In the caption to Figure 7B, the authors introduced ### as an indicator of statistical significance, but there is no such symbol in the figure itself. Also in the caption to Figure 7C, the authors enter the transcript # and ##, however, ### and ## are marked in the figure itself.

- In the caption to Figure 5, the authors should note the differences in the entered symbols of statistical differences * for IL-6 stained area, and # for DHE-stained area.

- It is necessary to indicate the dimension of the magnification scale on all representative images.

- In all captions to the figures, the authors should specify in what form they represent the data: mean ± SEM or mean ± SD.

6. Line 618. The sub-item number is erroneous. Not 2.10, but 3.10.

Author Response

Thank you for your valuable comments and suggestions. 

Please find attached doc as point-to-point response.

Reviewer 2 Report

Comments and Suggestions for Authors

Comments

Raising serum HDL-C levels and lowering TC and LDL-C is a good strategy to maintain healthy longevity by suppressing dyslipidemia and hypertension

Among the functional foods available on the market, policosanol (PCO) products are in great demand, having a number of beneficial properties preventing dyslipidemia and hypertension.

Due to the protective role of HDL in the body, it becomes important whether policosanol supplementation improves dyslipidemia, inflammation, organ damage and infertility with embryonal defects by increasing the quality of HDL-C and HDL. In this study was evaluated in vivo an efficacy of three policosanols from Cuba, China, and the USA in treating dyslipidemia and protecting the liver, kidney, ovary, and testis in a hyperlipidemic zebrafish model.

The results obtained in adult zebrafish and their offspring suggest that Cuban policosanol supplementation can rescue multiple organ dysfunction and infertility caused by HCD consumption, with the highest survival rate. At the same time, they may constitute a further contribution to conducting in vitro research.

I suggest publishing the manuscript in the presented version, but only after carefully examining the text and making linguistic corrections.

Comments on the Quality of English Language

re-reading the text and making minor corrections to the English

Author Response

Thank you for your kind words and encouragement.

We have meticulously reviewed the manuscript, addressing all typographical, grammatical, and linguistic errors. We hope that the revised manuscript meets the approval of the esteemed reviewer. 

Reviewer 3 Report

Comments and Suggestions for Authors

The study compared the effects of three policosanols from Cuba, China, and the USA on dyslipidemia and organ protection in hyperlipidemic zebrafish. The Cuban policosanol (Raydel®) demonstrated the highest survivability, improved lipid profiles, and organ protection. The study suggests potential benefits of policosanol consumption, emphasizing the importance of specific compositions in influencing efficacy.

1.      How do the specific compositions of aliphatic alcohols in different policosanols contribute to their varied efficacy in dyslipidemia and organ protection?

2.      Could the study explore the long-term effects and potential adverse reactions of policosanol supplementation in hyperlipidemic conditions?

3.      Are there variations in the bioavailability or metabolism of policosanols from different sources, and how might these factors influence their efficacy in vivo?

4.      The manuscript lacks a clear structure. Consider reorganizing the content into sections such as Introduction, Methods, Results, Discussion, and Conclusion for better readability.

5.      While the study design is robust, the introduction could provide more context on the relevance of zebrafish as a model for dyslipidemia and the specific significance of the chosen policosanols.

6.      The significance of the study is not well highlighted. Include a clear statement on how the findings contribute to the existing knowledge and potential implications for human health.

Author Response

(The authors gave the same response as above.)
